# Fecal Volatile Organic Compounds and Microbiota Associated with the Progression of Cognitive Impairment in Alzheimer’s Disease

**DOI:** 10.3390/ijms24010707

**Published:** 2022-12-31

**Authors:** Cristina Ubeda, María D. Vázquez-Carretero, Andrea Luque-Tirado, Rocío Ríos-Reina, Ricardo Rubio-Sánchez, Emilio Franco-Macías, Pablo García-Miranda, María L. Calonge, María J. Peral

**Affiliations:** 1Departamento de Nutrición y Bromatología, Facultad de Farmacia, Universidad de Sevilla, 41012 Sevilla, Spain; 2Departamento de Fisiología, Facultad de Farmacia, Universidad de Sevilla, 41012 Sevilla, Spain; 3Unidad de Memoria, Servicio de Neurología, Hospital Universitario Virgen del Rocío, 41013 Sevilla, Spain; 4Laboratorio de Análisis Clínicos, Hospital Universitario Valme, 41014 Sevilla, Spain

**Keywords:** Alzheimer’s disease, fecal volatile compounds, gut microbiota, metabolome, short-chain fatty acids, cognitive impairment

## Abstract

Metabolites produced by an altered gut microbiota might mediate the effects in the brain. Among metabolites, the fecal volatile organic compounds (VOCs) are considered to be potential biomarkers. In this study, we examined both the VOCs and bacterial taxa in the feces from healthy subjects and Alzheimer’s disease (AD) patients at early and middle stages. Remarkably, 29 fecal VOCs and 13 bacterial genera were differentiated from the healthy subjects and the AD patients. In general, higher amounts of acids and esters were found in in the feces of the AD patients and terpenes, sulfur compounds and aldehydes in the healthy subjects. At the early stage of AD, the most relevant VOCs with a higher abundance were short-chain fatty acids and their producing bacteria, *Faecalibacterium* and *Lachnoclostridium*. Coinciding with the development of dementia in the AD patients, parallel rises of heptanoic acid and *Peptococcus* were observed. At a more advanced stage of AD, the microbiota and volatiles shifted towards a profile in the feces with increases in hexanoic acid, *Ruminococcus* and *Blautia*. The most remarkable VOCs that were associated with the healthy subjects were 4-ethyl-phenol and dodecanol, together with their possible producers *Clostridium* and *Coprococcus*. Our results revealed a VOCs and microbiota crosstalk in AD development and their profiles in the feces were specific depending on the stage of AD. Additionally, some of the most significant fecal VOCs identified in our study could be used as potential biomarkers for the initiation and progression of AD.

## 1. Introduction

Alzheimer’s disease (AD) is a neurodegenerative disease characterized by a gradual loss of cognitive function and, ultimately, dementia [1]. AD starts with a non-symptomatic preclinical stage and progresses slowly in three clinical stages, i.e., early, middle and late. The cognitive deficit appears in the early stage and is accompanied by dementia through the middle and late stages. The early stage is considered incipient AD, also called mild cognitive impairment (MCI). According to the Global Deterioration Scale (GDS) developed by Reisberg, it corresponds to the stage GDS-3 [2]. While these patients are still independent and do not need assistance from family or caregivers, they show a subtle cognitive deficit but no dementia. In the middle stages, GDS-4 patients have a moderate cognitive decline and early dementia (mild AD). At GDS-5, patients have a moderately severe cognitive decline and moderate dementia (moderate AD) and they can no longer survive without assistance. The patients at the late stage exhibit advanced dementia (GDS-6 and GDS-7).

AD pathogenesis is characterized by an abnormal accumulation of the amyloid-β (Aβ) and hyperphosphorylated tau proteins in the brain that can be identified before the occurrence of cognitive decline and behavioral disturbances [3]. The method most frequently used to diagnose AD is based on the identification of the Aβ peptide and the tau proteins in the cerebrospinal fluid (CSF). Additionally, the Apolipoprotein E (APOE) alleles (ε2, ε3 and ε4) transmit different risks for the development of AD, with an increased risk with the ε4 allele [3].

Several genetic and environmental factors have been reported to be involved in the pathogenesis of AD. Among them, special attention has been given to the influence of a gut microbiota imbalance or dysbiosis in neurological health. Several reports from AD in animal models and patients have provided evidence of an association between the alterations of gut microbiota and AD pathogenesis via the gut-brain axis [4,5]. After the observation of specific trends in the gut microbiota composition with cognitive deterioration, some studies have postulated a critical role for the gut microbiome in the initiation and progression of the AD pathology. Therefore, the shifts in the gut microbiota composition might depict the mild cognitive impairment (MCI) and distinguish an early stage of AD from more advanced stages and from healthy subjects [6,7,8,9,10].

The effect of gut dysbiosis could be related to the metabolites generated by the bacteria that interact with the host physiology and may affect the functions of the intestine itself or the brain [11]. To detect, identify and quantify the changes in the metabolites in biological samples, several analytical techniques can be performed depending on their characteristics. Among the metabolites, the volatile organic compounds (VOCs) can be detected from several types of samples, including human breath, urine, sweat, blood and feces, using a non-invasive headspace sampling method and analysis based on a coupled gas chromatography-mass spectrometry. The VOCs may originate from both physiological and pathological processes in the human body and their levels of abundance can also reflect the health status of the organism [11]. The VOCs composition of human feces is the product of several factors such as the diet, microorganisms present in the intestinal tract both beneficial or eventually infectious and medical treatments [12]. The VOCs detected in the feces may reflect the gut microbiota composition and are considered to have potential use as biomarkers in the diagnoses of various diseases [11].

The goal of this work was to identify the volatile compounds in the feces from healthy subjects and AD patients using a headspace technique without the need for a complex sample preparation process. We aimed to associate the different profiles of the fecal VOCs and microbiota to AD, distinguishing the AD patients from the healthy subjects and, between the AD patients in the early stage before dementia developed in the middle stages. Additionally, to elucidate the relationship between the specific bacterial taxa and their metabolites, we performed a correlation analysis between the gut dysbiosis and the volatile compounds.

## 2. Results

### 2.1. Fecal Volatile Organic Compounds at the Early and Middle Stages of Alzheimer’s Disease

A total of 84 volatile organic compounds (VOCs) were determined in the feces of the participants of the present study by using HS-SPME/GC/MS (see Methods, Section 4.2). Among them, 21 alcohols, 21 terpenes, 12 aldehydes, 10 esters, 10 acids, 8 ketones and 2 sulfur compounds were identified (Appendix A). Figure 1 reveals that the feces of the AD patients accounted for the highest amount of VOCs compared to the healthy subjects. Therefore, these feces presented a higher volatile intensity. Moreover, differences were found in the volatile profile of each type of group. Therefore, the AD profile was mainly characterized by a greater abundance of acids and esters than that of the controls. Specifically, the most produced acids were short-chain fatty acids (SCFAs), propionic, acetic, butanoic and pentanoic acids and hexanoic acid (Appendix A). In addition, a slight increase in the alcohols, especially butanol, was detected in the AD patients. On the contrary, the quantity of terpenes, sulfur compounds and aldehydes was lower in the AD patients than in the healthy subjects.

Due to the high number of the determined volatile compounds, a multivariate analysis was used to determine the variables with a higher responsibility in the differentiation between the healthy subjects and the AD patients. First, the entire dataset was examined using a principal component analysis (PCA) to explore the data. As shown in Figure 2A,B, the scores and loadings plot displays three principal components (PCs) necessary to build a tridimensional plot. The projection of PC1, PC2 and PC5 grouped the samples according to the subject’s condition, revealing that the volatile profile of the feces was different between subjects with and without AD. Therefore, PC1 explained 16.14% and PC2 13.88% of the total variance. Although the percentage explanation of the variance was not very high, it must be considered that the fecal VOCs were the sum of multiple factors, as stated in the introduction. Therefore, the variance was expected to be distributed among these factors. Within the PCs, PC1 described the separation of the AD patient samples, placed on the positive side, from the controls, placed on the negative side (Figure 2A). Regarding the loadings plot (Figure 2B), this separation could be explained by a higher relationship between almost of the acids with the feces of the AD patients. Most of the terpenes were associated with the controls. The esters appeared in the AD patients in a higher amount than in the healthy subjects (Figure 1), especially propyl butanoate and butyl-2-methyl butanoate (Appendix A). Additionally, this tridimensional plot shows the grouping of the AD patients depending on the cognitive deterioration present in the patients of this study: Stage 3 (GDS-3: mild cognitive decline), Stage 4 (GDS-4: moderate cognitive decline), Stage 5 (GDS-5: moderately severe cognitive decline). This separation was the clearest when performing a PCA that focused only on the AD patients (Figure 3). The PCA showed that the differences between the volatile profile determined in the different cognition deterioration stages grouped the patients by stages in the plot with PC3 and PC4, explaining the 25.74% of the cumulative variance. PC4 described the separation of the cognitive deterioration degree, therefore, GDS-3 was placed on the positive side of PC4 and GDS-4 followed by GDS-5 were placed on the negative side of this axis. It is important to consider that the patients at GDS-3 showed the earliest subtle deficits but no dementia, however, GDS-4 patients had early dementia and GDS-5 patients had moderate dementia. Therefore, PC4 differentiated the patients between those with and without dementia. A significant relation between the acids with the early stage GDS-3 positioned only two of the 10 acids in the inferior half of the plane where the advanced stages of dementia were placed.

### 2.2. Most Relevant Fecal Volatile Organic Compounds at the Early and Middle Stages of Alzheimer’s Disease

Following the PCA, a partial least squares discriminant analysis (PLS-DA) (Appendix A) was performed, not for a classification approach due to the relatively low number of samples, but to examine whether the detected metabolites could be considered as significant candidates for AD biomarkers by the study of the variables with an importance in the prediction (VIP) (see Methods, Section 4.5). The selection of the VIPs was applied to reduce the number of the volatile compounds and to look for markers. From this selection, 29 VIP compounds were obtained (Appendix A). With the VOCs with a VIP value higher than 1, a heatmap was constructed to show the correlation of the total of 29 VIP compounds that were selected as the most relevant volatiles for the differentiation of the samples (Figure 4A). Of these, the potential biomarkers in the feces of the AD patients included 11 VOCs, namely the trans-2-decenal, propanoic, acetic, butanoic, pentanoic, hexanoic and heptanoic acids, butanol, 3-phenyl-propanol, butyl 2-methylbutanoate and propyl butanoate. In addition, the importance of some of these VOCs was confirmed by another statistic test, as shown in Figure 4B, with a volcano plot that obtained 15 VOCs from the 29 VIPs. This test enabled a quick visual identification of the VOCs that displayed large magnitude changes and were also statistically significant. The colored dots indicate the features that presented both a fold change >2 in the X-axis and the negative logarithm of the *p*-value < 0.05 in the Y-axis. The points laying to the right of the vertical line (in red) represent the VOCs whose value was greater in the AD patients than in the control subjects. Conversely, the points laying to the left side (in blue) represent those that are greater in the controls than in the AD patients. Among them, the trans-2-decenal and butanoic acids were considered to be the most significant in the differentiation between the controls and the AD patients, being positively correlated to the AD patients. On the contrary, dodecanol and benzaldehyde were highly correlated to the controls and 6-methyl-5-hepten-2-ol, 2-pentadecanone, dimethyl disulfide and 4-ethyl-phenol were the most remarkable volatiles in the feces of the healthy subjects (Figure 4). The high presence of terpenes in the control feces was noteworthy, showing the highest yields of gamma-bergamotene, beta bisabolene and humulene.

Next, we performed an analysis of variance (ANOVA) of the selected 29 VOCs (according to the VIPs), comparing the controls and AD patients grouped by stages. We only showed the compounds with differences that were statistically significant between the controls and the patients within a particular stage or between the patients within the different stages. Figure 5 reveals the significant increases in trans-2-decenal, propanoic, acetic, butanoic acids and butanol in the AD patients at GDS-3. Pentanoic (valeric) acid was increased in the GDS-3 and GDS-4 AD patients, heptanoic acid in GDS-4 and hexanoic acid in the GDS-4 and GDS-5 AD patients. On the other hand, benzaldehyde, dimethyl disulfide, 4-ethyl-phenol and dodecanol were significantly decreased in the AD patients at all stages and, therefore, were associated with the control subjects.

Taken together, these findings indicate that the healthy controls and the patients with AD exhibited a specific profile of the VOCs, and that the AD patients might be associated with the degree of cognitive deficit.

### 2.3. Fecal Volatile Organic Compounds and Life Style

In addition to these analyses, considering the potential influence of diet and lifestyle on the volatile composition of the feces of the participants, a dendrogram with a Spearman distance measure and a Ward clustering algorithm was obtained from the total VOCs identified for the AD patients and the control samples. The differences in the volatile profile due to the disease condition were higher than those acquired because of diet or lifestyle (Figure 6). Therefore, as can be observed, all the participants included in this study who were married couples (i.e., living in the same household and with the same diet), were highlighted with a different colored arrow and were not clustered, except for one. Therefore, the results obtained can be attributed to other reasons apart from their diet or lifestyle.

### 2.4. Gut Microbiota at Early and Middle Stages of Alzheimer’s Disease

The fecal volatile organic compounds (VOCs) profile may reflect changes in the gut microbiota composition. Therefore, we wondered whether there were changes in the microbiota and if any accompanied those that were observed in the VOCs with the progression of cognitive impairment in AD. The characterization of the gut microbiota was performed using 16S rRNA gene sequencing in the fecal samples (see Methods, Section 4.3), allowing us to identify the bacteria and quantify their relative abundance in the healthy controls and the AD patients that were previously classified in the stages GDS-3, GDS-4 and GDS-5.

The number of the total operational taxonomic units (OTUs) was 673 classified into 11 phyla, 17 classes, 30 orders, 73 families, 218 genera and 256 species. The total OTU richness (i.e., the number of the unique OTUs present in a subject) and the alpha diversity, measured with the Shannon index at the genus level, decreased in the AD patients at GDS-5 compared to all the groups (Figure 7A,B). The PCA revealed the differences in the genera depending on the cognitive decline degree and placed the AD patients grouped by the stages into the plot with PC2 and PC5 (Figure 7C). A total of 13 genera that differentiate the healthy controls from the AD patients were identified.

Next, we determined the contribution of each phylum in the microbiota composition. Figure 7D reveals that, in all groups of the participants, the *Firmicutes* exhibited the highest contribution followed by the *Bacteroidota* and *Actinobacteriota*. The *Firmicutes* contribution was lower and *Bacteroidota* was higher in the group of all the AD patients and in the stages GDS-4 and GDS-5. According to this, the ANOVA analysis of relative abundance (Figure 8A) shows that, compared either to the controls or the GDS-3 AD patients, the *Firmicutes* significantly decreased in GDS-4 and GDS-5 and the *Bacteroidota* and *Actinobacteriota* increased in GDS-4 and GDS-5, respectively. Within the *Firmicutes*, the class *Clostridia* was less abundant in the patients at GDS-4 and GDS-5 and the family *Veillonellaceae* at GDS-5 (Figure 8B). At the genus level, compared to the control, we found rises in *Faecalibacterium*, *Roseburia* and *Lachnoclostridium* in the AD patients at GDS-3, in *Peptococcus* at GDS-4, in *Ruminococcus* and *Blautia* at GDS-5 and diminutions in *Clostridium* and *Coprococcus* in all the studied stages (Figure 8C).

Within the phylum *Bacteroidota*, compared to the control and the GDS-3 AD patients, the class *Bacteroidia* and the genus *Barnesiella* were more abundant in the GDS-4 patients and the family *Bacteroidacea* and the genera *Bacteroides* and *Alistipes* increased in the GDS-4 and GDS-5 patients (Figure 9A). Regarding the phylum *Actinobacteriota*, compared to all the other groups, we observed an increase in the abundance of the class Actinobacteria in the GDS-5 patients but a reduction in the family *Bifidobacteria* and the genus *Bifidobacterium* at GDS-3 and GDS-4 (Figure 9B). In addition, within the phylum *Verrucomicrobiota*, the family *Akkermansiaceae*, the genus *Akkermansia* and the specie *Akkermansia muciniphila* were more abundant in the GDS-3 AD patients compared to the other groups (Figure 9C).

Additionally, we also found significant differences in the OTUs when compared between the control subjects and the AD patients (Appendix A).

In summary, the alterations in the gut microbiota in the AD patients were more significant in the patients at GDS-5 who exhibited moderate AD but with greater cognitive deficit and dementia than the GDS-4 or GDS-3 patients.

### 2.5. Correlations between the Fecal Volatile Organic Compounds and the Gut Microbiota in the Healthy Subjects and the Alzheimer’s Disease Patients

To identify the associations between the most relevant fecal VOCs and the 13 bacterial genera abundances, the correlation coefficients and their statistical significance were calculated using Microsoft Excel. Figure 10 reveals that all the correlations were positive and were most significant between the VOCs and the genera from the phylum *Firmicutes*. The VIP volatile compounds associated with AD that exhibited significant correlations with the bacterial taxa were propyl butanoate, butyl 2-methylbutanoate, butanol, acetic, propanoic and butanoic acids with *Lachnoclostridium*, butanol with *Roseburia* and *Akkermansia* and pentanoic acid with *Faecalibacterium*. Additionally, other significant volatile acids, such as hexanoic acid, were correlated with both *Ruminococcus* and *Bifidobacterium*, and heptanoic acid with *Peptococcus*. In contrast, the VIP volatile compounds associated with the healthy subjects that displayed significant correlations with the bacterial taxa were: 4-ethyl-phenol with both *Clostridium* and *Coprococcus*, dodecanol with *Clostridium*, dodecanal with *Coprococcus*, gamma-bergamotene with *Coprococcus,* and beta-bisabolene with *Clostridium*.

These findings agreed with some of the ANOVA analysis (Figure 5, Figure 8 and Figure 9), showing the changes in the same direction specifically in a VOC and a bacterial taxon in the same AD stage compared to the controls. Therefore, butanol, acetic, propanoic, butanoic, pentanoic acids, *Lachnoclostridium*, *Roseburia*, *Faecalibacterium* and *Akkermansia* increased in the AD patients at GDS-3, heptanoic acid and *Peptococcus* increased in the AD patients at GDS-4 and hexanoic acid and *Ruminococcus* increased in the AD patients at GDS-5. The abundances of 4-ethyl-phenol, dodecanol and *Clostridium* and *Coprococcus* were decreased in all the studied stages of AD and, therefore, were associated with the heathy subjects.

### 2.6. Fecal Calprotectin in the Alzheimer’s Disease Patients

Since we observed an increase in the abundance of the genera *Ruminococcus* and *Blautia* in the AD patients at GDS-5, and that increases of them have been previously reported in intestinal inflammation, we wondered whether they also displayed gut inflammation. For this study, we measured the levels of the fecal calprotectin, a marker of intestinal inflammation, using the Particle-Enhanced Turbidimetric Immunoassay (PETIA) (see Methods, Section 4.4). Figure 11 reveals the data as the mean ± the SEM, showing a slight rise in the AD patients, especially at GDS-5. However, it was not significant compared to the control subjects. These findings suggest a trend toward gut inflammation in AD.

## 3. Discussion

There is an important influence of an imbalance in the gut microbiota, probably via the metabolites production, in neurological health. Metabolic changes are the quickest indicator of abnormalities occurring in the organism due to a high dynamism of metabolite homeostasis [13]. Among the metabolites generated by microbiota, the VOCs may be involved in the physiological or pathological mechanisms and represent a potential biomarker for the diagnosis and the progression of diseases. In this work, the fecal samples obtained from the healthy subjects and the AD patients at the early and middle stages were used to look for the VOC profiles. Since the VOCs extraction procedure can affect the results, a method with minimal manipulation was employed, i.e., placing the sample in a vial and analyzing the headspace generated.

The general volatile profile of the feces of the healthy controls *versus* the AD patients was quite different, firstly since the higher abundance of the volatiles detected in the AD mainly attributed to the larger presence of acids found in the AD patients. This significant presence of acids points to the microorganisms as plausibly responsible. Their metabolic abilities enable the transformation of the complex nutrients into the short-chain fatty acids (SCFAs), mainly acetic, propionic and butyric acids [14]. The SCFAs are the main metabolites of the gut microbiota formed during the fermentation of non-digestible carbohydrates [15]. Recent studies have described an association between an altered gut microbiota derived from the SCFAs and the pathogenesis of AD [16]. Clinical and preclinical studies have also shown changes in the SCFAs with AD (Table 1).

In our study, the SCFAs (acetic, propanoic, butanoic and pentanoic acids) joined to hexanoic and heptanoic acid yielded the highest amounts in the AD patients. These results agreed with those observed in the AD mice models in which either the SCFAs, such as acetic, butanoic and propanoic acids [17] or the fecal propionate [18] were highly present in the feces. Conversely, these results partially disagreed with Zheng et al. [19] who found that propanoic, isobutyric and 3-hydroxybutyric acids in the feces of an AD mice model were lower in quantity but more complex acids, such as 2-hydroxy isobutyric, levulinic and valpronic acids, were increased. The more advanced stage of the AD mice model employed by Zheng et al. [19] might explain the different results observed by Favero et al. [17] and by us at the early stage of AD. Therefore, among the different degrees of AD that were studied, the early stage (GDS-3) was highly related to the SCFAs while they were significantly less related to the more advanced stages (GDS-4 and GDS-5). Recently, progressive decreases in the fecal SCFAs, but not hexanoic acid, have been associated with cognitive decline, suggesting that these SCFAs could be used as AD biomarkers for the transformation from mild cognitive impairment (MCI) to more advanced AD, as we observed [10]. In another study, no differences were found in the fecal propionate and butyrate in MCI compared to the normal subjects [7]. In an AD mice model, only the fecal butyric acid tended to be lower [20]. The SCFAs (especially butyric and propanoic acids) have beneficial effects on the intestine, due to their capacity to maintain metabolism, barrier integrity, immune homeostasis and exhibit anti-inflammatory effects [21]. In the brain, the SFCAs maintain the blood-brain barrier (BBB), regulate the metabolism, promote neurotransmitters synthesis and interfere with amyloid formation [22,23,24]. According to the neuroprotective effects of the SCFAs, it has been observed that the restoration of the intestinal butyrate improved memory in an AD mice model [25] and that donepezil administration, commonly used for the treatment of mild or moderate AD, increased the butanoic acid in the feces from an AD mice model [26]. Other studies in AD mice models showed increases in propionate in the brain of AD mice and that the supplementation of the SCFAs promoted microglial activation and could increase amyloid deposition [16,27]. Marizzoni et al. [28] reported that acetate and valerate (pentanoate) in the blood were positively associated with an amyloid deposit in the AD patients. The SCFAs might be crucial mediators of the gut microbiota in AD pathogenesis [10,16]. The fecal SCFAs augment observed in our GDS-3 AD patients could point to a microbiota response at the beginning of AD to produce gut-brain self-protective molecules. These molecules could decrease their effectiveness with an increase in the dysbiosis, although on the other hand they could induce the onset of AD dementia. Depending on the type and concentration of the SCFAs, their effects are different, and the inconsistency of the available studies indicates that the role of the SCFAs in the AD pathogenesis is rather complicated and demands further investigation [16]. Apart from the SCFAs, butanol was more abundant in the AD patients and had a strong relation between the early stage GDS-3 and the controls. This alcohol was previously described as the key to distinguish Parkinson’s disease in patients with Alzheimer’s disease and in healthy individuals by measuring exhaled breath using an electronic nose [29].

**Table 1 ijms-24-00707-t001:** Investigation of the SCFAs in the AD patients and in the animal models of AD.

Subjects	Number of Subjects	Sex	Reference
AD patients	17	5 Males/12 Females	[7]
AD patients	77	38 Males/39 Females	[10]
APP/PS1 mice 3×Tg-AD mice	30 8	13 Males/17 Females	[16]
		n.i.	
3×Tg-AD mice	4	n.i.	[17]
3×Tg-AD mice	16	0 Males/16 Females	[18]
APP/PS1 mice	56	n.i.	[19]
APP/PS1 mice	6	6 Males/0 Females	[20]
APP/PS1 mice	8	n.i.	[25]
SAMP-8 mice	24	24 Males/0 Females	[26]
3×Tg-AD mice	8	0 Males/8 Females	[27]
AD patients	89	41 Males/48 Females	[28]

APP/PS1: amyloid precursor protein/presenilin 1, 3×Tg-AD (triple transgenic-Alzheimer’s disease), SAMP-8: senescence-accelerated prone-8. n.i.: not provided information.

Furthermore, the healthy subjects showed higher amounts of dodecanol compared to the AD patients, the clearest best biomarker of the health condition. It was found that the administration of the neuroprotectors that alleviated the cognitive impairment in an AD mice model was reflected, among other compounds, in an increase in dodecanol in urine samples [30]. In addition, there was a higher abundance in the aldehydes in the feces of the healthy subjects as well as benzaldehyde and dodecanal. A higher presence of aldehydes was also observed in the urine of the healthy mice compared to an AD mice model [26]. Nevertheless, one of the aldehydes determined in our study was firmly positioned as a potential AD biomarker for early stage (GDS-3) detection, namely trans-2-decenal. This aldehyde was related to a growth-inhibitory effect on the pathogen bacteria and fungi and showed protective effects on the beneficial bacteria, alongside action against plant parasites [31,32,33]. The gut microbiota could display some type of defense mechanism against the foreign bacteria and fungi colonization through this compound or even through new protective microbiota species that produce this compound. Other compounds associated to the healthy controls were the terpenes. These compounds, usually found in nature such as plants, fungi, insects, marine organisms, etc., provide a typical floral and citric aroma. They play a very important role in human health and demonstrated a variety of anticancer, antimicrobial, anti-inflammatory, antioxidant and immunomodulatory activities, among others [34]. Particularly, their neuroprotective effect in the AD animal models has been widely tested [35,36,37,38,39]. It can be hypothesized that the gut microbiota of the healthy subjects has the enzymatical ability to liberate terpenes from ingested food. Therefore, a healthy gut microbiota could exert protective effects after being reabsorbed in the intestine.

Benzaldehyde, dimethyl disulfide 4-ethyl-phenol and dodecanol were significantly decreased in our AD patients at all stages and, therefore, related to the controls. Sulfur compounds and dimethyl disulfide were claimed to be positive against amyloid-β-induced neurotoxicity and cognitive impairment in mice [40]. The higher presence of 4-ethyl-phenol in the healthy mice compared to the AD mice was also observed by Tian et al. [26] in urine, but not in feces. This compound was found to increase in the brain and liver of an AD mice model [41]. The low presence of this VOC in the feces of the AD patients might reveal possible locations in other areas of the body.

The increased presence of esters was expected since a higher proportion of acids and alcohols were determined in the feces of these patients and that their esterification through several bacteria metabolism would give rise to esters. Tian et al. [26] also observed a higher content of esters, especially methyl esters, in the feces of an AD mice model.

Since the fecal VOCs profile may reflect the changes in the gut microbiota composition, the microbiota in the fecal samples was analyzed. According to previous reports [42], a decreased gut microbial richness and diversity were found in the AD patients with moderate AD (GDS-5) but not in the patients with a lower cognitive impairment (GDS-3 and GDS-4), indicating a trend toward a progressive microbiota alteration from an early stage to middle AD. A lower abundance of the phylum *Firmicutes* was observed in the AD patients at GDS-4 and GDS-5 who already showed stages of dementia. Within *Firmicutes*, the genera *Clostridium* and *Coprococcus* decreased in the three stages studied and, therefore, their presence was related to the healthy subjects. Recent reports have also shown reductions in these bacteria taxa in the AD patients [6,8,43,44]. We also showed positive correlations that suggest that, in healthy subjects, *Clostridium* could produce 4-ethyl-phenol, dodecanol and beta-bisabolene, and *Coprococcus* could produce 4-ethyl-phenol, dodecanal and gamma-bergamotene. The production of 4-ethyl-phenol from tyrosine has been attributed to various species of *Clostridium* and that of dodecanal to *Coprococcus* [45,46]. In other genera of *Firmicutes*, we found increases in their abundance depending on the degree of the cognitive deficit, agreeing with the previous reports obtained in the AD patients [6,8,9,47,48]. Additionally, the positive correlations together with the ANOVA analysis suggested the VOCs that could be produced by these *Firmicutes* genera and in which stage of AD. Therefore, in the early stage (GDS-3), *Faecalibacterium* was associated with pentanoic acid and butanol, *Roseburia* with butanol and *Lachnoclostridium* with acetic, propanoic and butanoic acids and butanol. Interestingly, in the middle stages, coinciding with dementia development, *Peptococcus* was related to heptanoic acid at GDS-4 and *Ruminococcus* with hexanoic acid at GDS-5. *Faecalibacterium* and *Lachnoclostridium* are known producers of the SCFAs [45,49].

Concerning the phylum *Bacteroidota*, its enrichment in our AD patients was reflected in the genera *Bacteroides* and *Alistipes* that were more abundant in the GDS-4 and GDS-5 patients, which was in line with previous findings [6,50,51]. However, other reports observed that members of *Bacteroidota* were more abundant in MCI [8] or less abundant in the AD patients [47,48]. Within the phylum *Actinobacteriota*, the genus *Bifidobacterium* was less abundant in the GDS-3 and GDS-4 AD patients, as previously described [6]. In addition, we identified increases in *Akkermansia* (phylum *Verrucomicrobiota*) in our AD patients at GDS-3, as seen previously in the patients with MCI [48], although an *Akkermansia* increase in the AD patients with dementia was also described [6,44,51]. Our correlations indicated that *Bifidobacterium* was associated with hexanoic acid and *Akkermansia* with butanol. The abundance of *Akkermansia muciniphila*, known by its intestinal anti-inflammatory effect [52], was higher in our AD patients at GDS-3. *Roseburia* and *Faecalibacterium*, known for their anti-inflammatory effects in the intestine through the SCFAs [53], were also more abundant in the feces of our GDS-3 AD patients. On the other hand, in the GDS-5 AD patients, we found increases in the genera *Ruminococcus* and *Blautia* that were also elevated in chronic intestinal inflammation [54]. *Blautia* has been related to more advanced AD [6]. However, there is some discrepancy in the association of this genus with diseases and most of the properties are related with potential probiotic functions [55]. We also found that the fecal calprotectin levels tended to increase in the AD patients, especially at GDS-5, suggesting possible intestinal inflammation. Leblhuber et al. [56] reported fecal calprotectin levels in the AD patients at higher-than-normal values. In the AD patients, the pro-inflammatory gut bacterial taxa were more abundant and the anti-inflammatory taxa were less abundant [57]. Gut dysbiosis and intestinal inflammation are directly associated with the gut barrier dysfunction that may contribute to the AD pathology [58]. The gut barrier disruption could be promoted by a decrease in the protective effect of the SCFAs, making the barrier more permeable which would favor the absorption of harmful molecules into the bloodstream or allow the passage of bacteria and their metabolites that might induce inflammation or the initiation of a cancer-fighting immune response [59,60]. Since AD is most common in elderly people, age-related oral health problems could be considered as triggers for oral microbiota disturbances [61]. This altered microbiota might contribute to intestinal inflammation by inducing gut dysbiosis due to the swallowed oral pathobionts [62].

Remarkably, different patterns in the gut microbiota were linked to the stages of AD, therefore, depending on the cognitive decline and dementia development. A recent systematic review concluded that the AD patients exhibited an altered gut microbiota, and the differences were mediated by the clinical stages of the disease [42]. Li et al. [48] did not find any differences in the gut microbiota between the predementia stage, MCI, compared to the AD patients with more advanced stages. Other studies showed different shifts in the gut microbiota associated to the predementia stage or to the AD dementia stages compared to the healthy subjects [7,8,10,48]. None of these profiles exactly coincided with those of our AD patients in the studied stages, as the gut metabolome would be different in each case. To our knowledge, two reports investigated the metabolic output of the altered gut microbiota in the AD patients, obtaining the associations between the bacterial taxa and the metabolites, most of which were non-volatile [9,10] and only five of which were identified as volatile SCFAs and associated with AD progression [10].

Although our results are promising and statistically significant, the study presented some limitations since the number of the AD patients for each stage was low and the number of males and females was unbalanced in the control group. In addition, the AD patients received medication that could affect the gut microbiota composition. This study was a first approach to help elucidate the relationship between the changes in the volatile compounds and those of the gut microbiota that occur in the feces as the AD progresses. Further studies in large cohorts are required to obtain consistent associations for each stage of AD.

## 4. Materials and Methods

### 4.1. Participants

The study was approved by the Ethics Committee of Virgen Macarena and Virgen del Rocío University Hospitals from the Junta de Andalucía (20199694153_S1900169). Informed consent was obtained from all the subjects involved in the study. The participants, the Alzheimer’s disease (AD) patients and the control subjects (cognitively normal) were recruited between September 2020 and January 2021 at the Virgen del Rocío University Hospital, Sevilla, Spain. The healthy subjects (controls) were cognitively normal as indicated by the evaluation carried out in parallel with the AD patients. The AD patients were staged according to the Global Deterioration Scale (GDS) and the functional assessment staging (FAST), also known as Reisberg scale [2]. The GDS assesses the status of a patient’s primary dementia, regarding the progression of their disease, and defines the stages of cognitive decline. The patients are classified in seven stages (1–7): Stage 1 (GDS-1: no cognitive decline), Stage 2 (GDS-2: very mild cognitive decline), Stage 3 (GDS-3: mild cognitive impairment (MCI), no dementia), Stage 4 (GDS-4: moderate cognitive decline, early dementia), Stage 5 (GDS-5: moderately severe cognitive decline, moderate dementia), Stages 6 and 7 (GDS-6 and GDS-7: severe cognitive decline, advanced dementia) [2,63].

The participants included in the study included ten cognitively normal control subjects (8 women, 2 men) and twelve AD patients (6 women, 6 men). Of the AD patients, four were at GDS-3, four at GDS-4 and four at GDS-5 with two women and two men per group. The participants did not differ in age (70 ± 3.5 for GDS-3 group, 62 ± 2.6 for GDS-4, 66 ± 3 for GDS-5 and 66 ± 3 for the control group). Among the participants were four married couples between the control subjects and the AD patients.

The diagnostic of AD was realized by the detection of specific biomarkers such as Aβ42 or the total (T-tau) and phosphorylated tau (P-tau) proteins in the cerebrospinal fluid, or the Apolipoprotein E (APOE) ε3/ε4 genotype or fluorodeoxyglucose-positron emission tomography (FDG-PET).

All the participants were on an omnivorous diet and none of them reported any dietary restrictions or an intake of antibiotics, anti-inflammatory, probiotic or prebiotic products within the preceding 3 months. None of the subjects included in the study had a history of gastrointestinal disorders.

Concerning the AD patient medication, all the patients at GDS-5 were taking donepezil, half of the patients at GDS-4 were taking memantine, half of the patients at GDS-3 were taking a lower dose donepezil and the other half of the GDS-3 patients did not receive any medication. The rest of the patients had not taken these medications.

Regarding sample collection, the first morning fecal samples were collected by all the participants involved in the present study in sterile fecal collection containers at home and packaged with frozen gel packs. Upon receipt, the samples were immediately aliquoted into sterile tubes and frozen at −80 °C until further analysis. The fecal samples were processed depending on the analysis (see Methods, Section 4.2, Section 4.3 and Section 4.4).

### 4.2. Analysis of Volatile Organic Compounds by HS-SPME/GC/MS

Headspace-solid phase microextraction/gas chromatography/mass spectrometry (HS-SPME/GC/MS) were performed to analyze the volatile organic compounds (VOCs). Approximately 3 g of the feces were placed in a 20 mL solid phase microextraction (SPME) vial and mixed with 3 mL of a sodium chloride solution (20%) by vortexing for 1 min to help the volatile compounds displace to the headspace of the vial. Subsequently, 10 μL of 4-methyl-2-pentanol (0.75 g/L) was added as an internal standard. For this purpose, an Agilent 88900 gas chromatograph coupled to an Agilent 5977B simple quadrupole mass spectrometer (Agilent, Santa Clara, CA, USA) equipped with a multipurpose autosampler MPS Robotic Pro LS (Gerstel, Müllheim an der Ruhr, Germany) and with a thermostatic tray at 20 °C was used. For the extraction of the VOCs from the feces, a 2 cm SPME fibre 50/30 μm DVD/CAR/PDMS (Supelco, Bellefonte, PA, USA) was employed. The samples were incubated at 45 °C for 5 min at 300 rpm. Then, the fibre was exposed to the vial headspace for 30 min. Afterwards, the desorption of the compounds adsorbed into the fibre was performed for 3 min in the injection port at 250 °C in the splitless mode at 54 mL/min. The column used for the separation of the compounds was a J&W CPWax-57CB 50 m × 0.25 mm and 0.20 μm film thickness (Agilent, Santa Clara, CA, USA) with helium as the carrier gas at 1 mL/min. The oven temperature program was as follows. It started at 35 °C for 3 min increasing to 4 °C/min to 220 °C, which was then held for 1 min. The volatile compounds were detected in the full scan mode at 70 eV and a mass registration from 29 to 300 *m*/*z*. All the data were recorded using the MS ChemStation. The compound identification was based on mass spectra matching using the 2.0 version of the standard NIST library and the linear retention index (LRI) of the authentic reference standards from literature. The data were expressed in a relative peak area with respect to the internal standard (4-methyl-2-pentanol) and normalized dividing the peak area of the target ion of each compound by the peak area of the target ion of the internal standard. The linear retention index of each compound was calculated by the retention times of n-alkanes (C6-C30) under identical conditions for each analysis program.

### 4.3. Gut Microbiota Analysis

The gut microbiota analysis of the fecal samples from the AD patients and the control subjects was performed through the AllGenetics laboratories (AllGenetics & Biology, S.L., A Coruña, Spain) by using 16S ribosomal RNA (rRNA) gene sequencing. First, the DNA from the fecal samples (0.25 g) was extracted using the QIAamp PowerFecal Pro DNA isolation kit (Qiagen, Madrid, Spain), following the manufacturer’s instructions. For the library preparation, a fragment of the bacterial 16S rRNA region (300 bp) was amplified using the primers 515F-Y (5′ GTG YCA GCM GCC GCG GTA A 3′) and 806R (5′ GGA CTA CNV GGG TWT CTA AT 3′). The libraries were purified using the Mag-Bind RXNPure Plus magnetic beads (Omega Biotek, Norcross, Georgia, USA) and sequenced in a NovaSeq PE250 (Illumina, San Diego, CA, USA). The obtained 16S amplicon reads were processed using a QIIME2 microbiome bioinformatics platform [64]. The Divisive Amplicon Denoising Algorithm 2, implemented in QIIME2, was used for processing the sequencing data and clustering the resulting sequences into the amplicon sequence variants (ASVs). The taxonomy was assigned to the ASVs using the feature-classifier classify-sklearn approach, a plugin of QIIME2. Additionally, the non-bacterial ASVs such as the eukaryotic sequences of chloroplast and mitochondrial origin were removed from the final ASV and they were not used for the downstream analyses. The final filtered ASV table was converted into a Biological Observation Matrix file (.biom), which was directly imported into the R version 3.6.1 using the package phyloseq to plot the results of the analyses.

### 4.4. Fecal Calprotectin Quantification

The fecal calprotectin was quantified by a Particle-Enhanced Turbidimetric Immunoassay (PETIA) with the BÜHLMANN fCAL^®^ turbo test (BÜHLMANN Laboratories AG, Schönenbuch, Switzerland) on the Roche cobas^®^ c 501 clinical chemistry analyzer (Roche Diagnostics, Basel, Switzerland). For this study, 1 g of feces from each participant was used. The fecal extracts were prepared using the CALEX^®^ Cap device kit following the manufacturer’s instructions and achieving a final dilution of 1:500 with an extraction buffer. The fecal extracts were incubated with the saline reaction buffer MOPS (3-(N-morpholino) propanesulfonic acid) and mixed with the polystyrene beads coated with avian antibodies against human calprotectin (immunoparticles). The sample turbidity increased with the calprotectin-immunoparticle complex formation and was proportional to calprotectin concentration. The detected light absorbance allowed for the calprotectin quantification by interpolation on a validated calibration curve with the controls. The results were automatically calculated on the clinical chemistry analyzer (Roche Diagnostics, Basel, Switzerland) and expressed in μg of calprotectin/g of the feces. The cut-off value was set at 50 μg/g (normal), values 50–200 μg/g represented the grey-zone, but the inflammation could not be excluded and >200 μg/g was indicative of active intestinal inflammation.

### 4.5. Statistical Analysis

The multivariate analyses of the volatile organic compounds (VOCs) were performed using MetaboAnalyst 5.0 (Xia Lab, McGillUniversity, Montreal, QC, USA). The hierarchical clustering and heatmap were constructed using the Pearson correlation coefficient based on the relative peak areas of the extracted VOCs selected by their VIP value > 1 to determine the most relevant samplesin each group of participants. The volcano plot, which combined the results from the fold change analysis and *t*-tests into one single graph, was also performed in this case with the total VOCs determined to visually identify the significant features based on their statistical significance. The X-axis plotted the fold change between the two groups (on a log scale), while the Y-axis represented the *p*-value for a *t*-test of the differences between the samples (on a negative log scale). The goal of the fold change analysis was to compare the absolute values of change between the two group means. The significant features were those features whose FCs were beyond the given FC threshold (either up or down). Prior the modeling, the data were autoscaled.

The principal component analysis (PCA) and a partial least squares-discriminant analysis (PLS-DA) were developed using PLS-Toolbox version 7.9.5 (Eigenvector Research Inc., Wenatchee, WA, USA) working under a MATLAB environment (Matlab R2016a; The MathWorks, Inc., Natick, MA, USA). Prior the modeling, the data were autoscaled. The PCA was carried out with the relative peak area of the total VOCs to explore the data and ascertain the degree of differentiation between the samples. The PLS-DA was performed for the feature selection to examine which VOCs were relevant for the differentiation. For that, one of the feature importance measures commonly used in PLS-DA was the variable importance in projection (VIP) score, which was calculated to estimate the importance of each variable. It was a weighted sum of the squares of the PLS loadings, which were based on the amount of the explained Y-variance in each dimension. Therefore, it measured the importance of each variable in the PLS-DA model that summarized the contribution of a variable to the model. The VIP scores were scaled so that all the predictors with a VIP greater than one were considered relevant, with a higher relevance corresponding to a higher VIP [65]. The cross-validation was applied to validate the model (venetian blind, five splits).

For the gut microbiota analysis, the alpha diversity was calculated by measuring the Shannon index using the R version 3.6.3 with the package vegan. The ANOVA analysis of the richness, alpha diversity, relative abundance of the gut microbiota and relative abundance of the VOCs was performed using the GraphPad Prism software, version 8.0. The data are presented as the mean ± the standard error of the mean (SEM). The exact n, i.e., the number of participants, was reported in the figure legends. The comparisons between the groups were evaluated with the ANOVA followed by the Tukey post hoc test. The differences were set to be significant at *p* < 0.05. The principal component analysis (PCA) was carried out with the area of the relative abundance of the gut microbiota at the genus level to differentiate between the stages of AD by using PLS-Toolbox version 7.9.5 (Eigenvector Research Inc., Wenatchee, WA, USA) working under a MATLAB environment (Matlab R2016a; The MathWorks, Inc., Natick, MA, USA).

The correlations between the selected VOCs by the VIP scores, the bacterial genera abundance and the significances determined using the *t*-test were carried out using the data analysis package from Microsoft Excel.

## 5. Conclusions

In the present study, for the first time, a high number of volatile compounds belonging to seven different chemical families were identified in the feces of the healthy controls and in those of the AD patients classified in the early and middle stages. Our results revealed different patterns of gut dysbiosis that might generate different VOC profiles distinguishing the early stage without dementia from the middle stages with dementia. The early stage (GDS-3) of AD is characterized mainly by a higher abundance in the SCFAs and their producing bacteria, *Faecalibacterium* and *Lachnoclostridium*. Interestingly, coinciding with the dementia development in the AD patients, parallel increases in heptanoic acid and *Peptococcus* were observed. At a more advanced stage of AD, the gut microbiota and volatiles shifted towards a profile with increases in hexanoic acid, *Ruminococcus* and *Blautia*. These modifications could be linked with the progressive cognitive decline and associated with AD development and the onset of dementia. Additionally, the findings suggested that the most relevant fecal VOCs identified could be potential biomarkers for the initiation and progression of AD. Further studies are required to associate the biomarkers with each stage of AD.

## Figures and Tables

**Figure 1 ijms-24-00707-f001:**
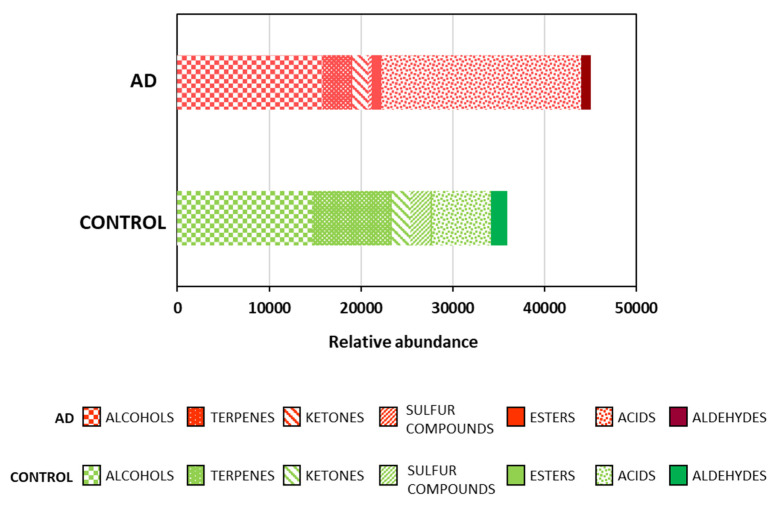
Relative abundance of the volatile organic compounds (VOCs) in the feces from the control subjects and the Alzheimer’s disease (AD) patients; (n = 10, control subjects; n = 12, AD patients).

**Figure 2 ijms-24-00707-f002:**
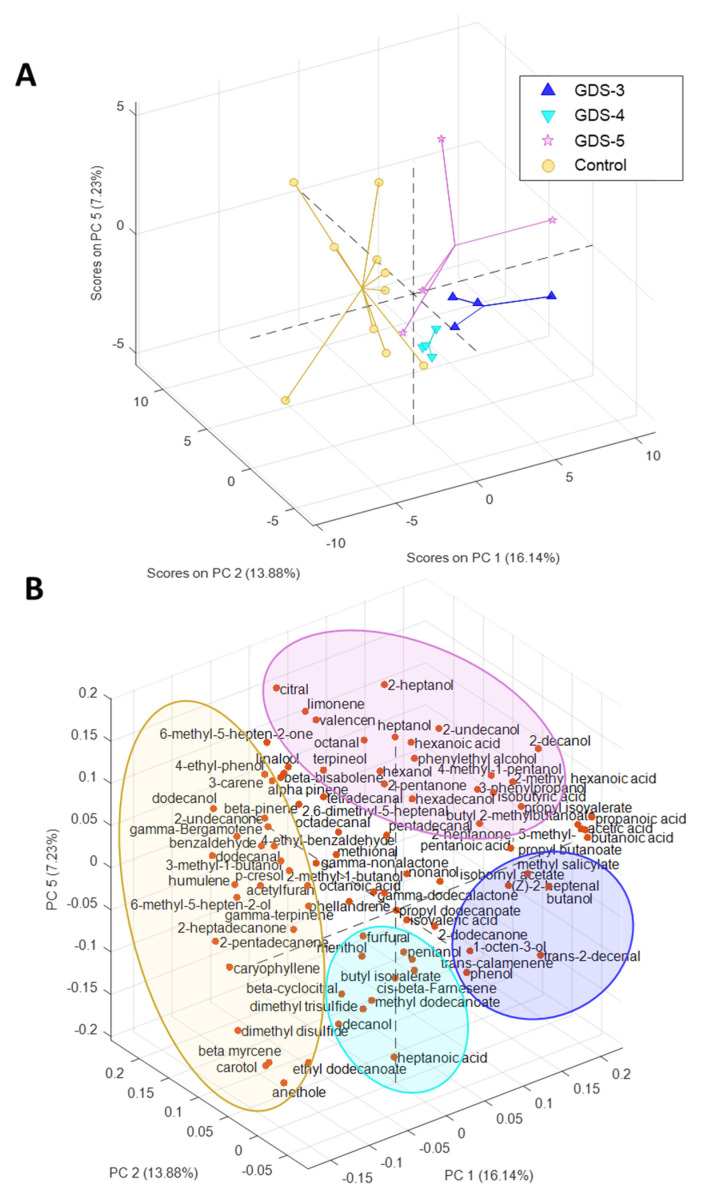
Principal component analysis (PCA) of the fecal volatile organic compounds (VOCs) in Alzheimer’s disease (AD) and the control subjects. The (**A**) scores and (**B**) loadings plots of the PCA model carried out with the relative peak area of the total VOCs obtained from the control subjects and the AD patients classified in the stages: GDS-3, GDS-4 and GDS-5; (n = 10, control subjects; n = 12, AD patients; 4 per stage).

**Figure 3 ijms-24-00707-f003:**
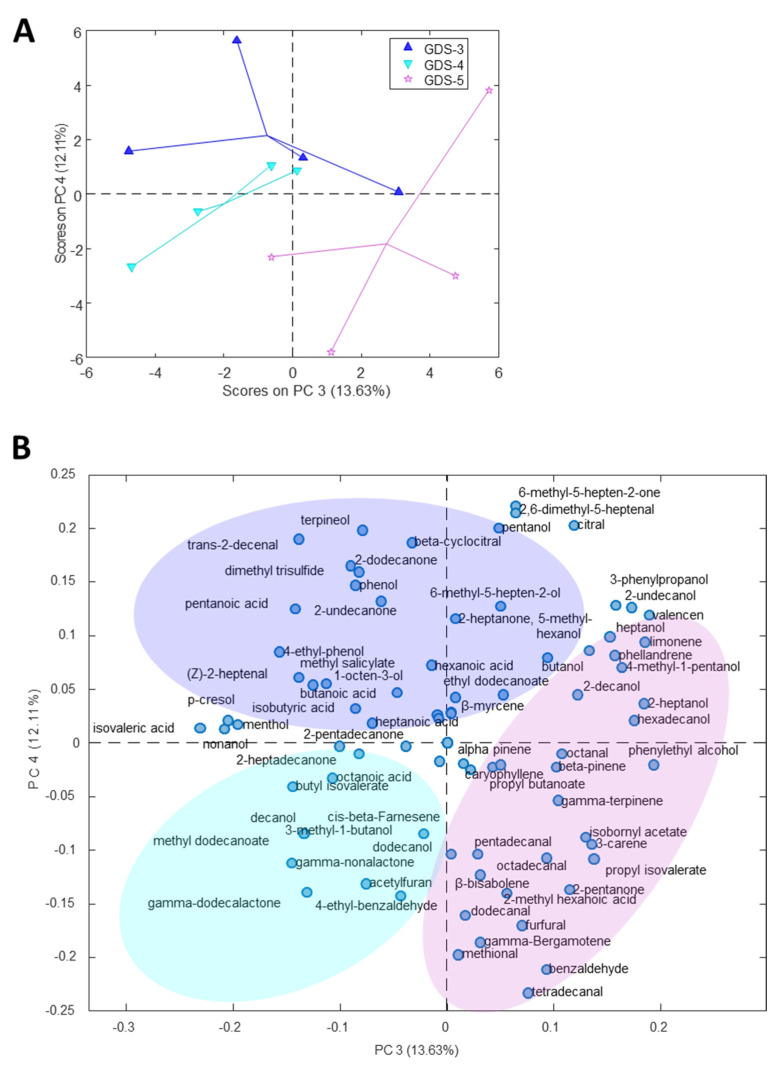
Principal component analysis (PCA) of the fecal volatile organic compounds (VOCs) in Alzheimer’s disease (AD). The (**A**) scores and (**B**) loadings plots of the PCA model developed with the fecal samples from the AD patients and the relative peak area of the total VOCs identified; (n = 12, AD patients; 4 per stage).

**Figure 4 ijms-24-00707-f004:**
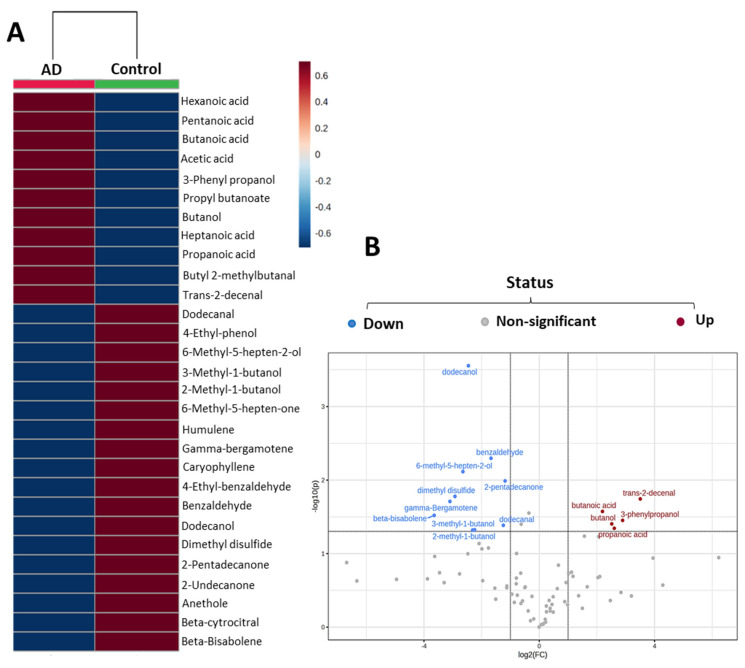
Most relevant fecal volatile organic compounds (VOCs) in the control subjects and the Alzheimer’s disease (AD) patients. (**A**) Heatmap plot of the 29 most relevant VOCs selected by the VIP scores. (**B**) Volcano plot developed with the relative peak area of the total VOCs identified. The more significant VOCs are colored in blue (down) and in red (up); (n = 10, control subjects; n = 12, AD patients; 4 per stage).

**Figure 5 ijms-24-00707-f005:**
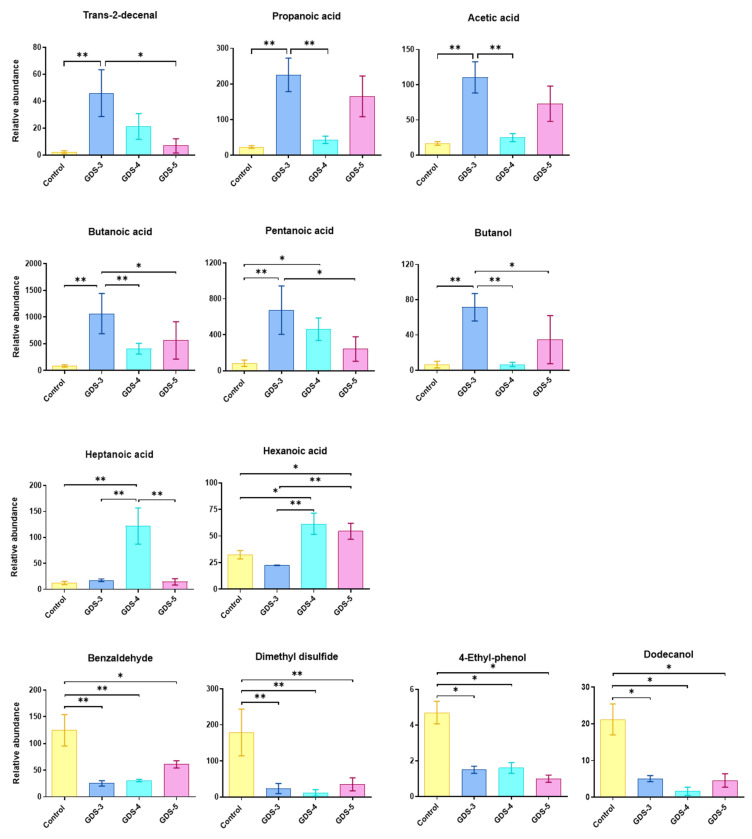
Fecal volatile organic compounds (VOCs) selected by the VIP scores in the control subjects and the Alzheimer’s disease (AD) patients who showed significant differences between the control and/or the AD stages. The VOCs relative abundance in the feces from the control subjects and AD patients were classified in stages GDS-3, GDS-4 and GDS-5. The data are the means ± the SEM, (n = 10, control subjects; n = 12, AD patients; 4 per stage). The ANOVA showed an effect (*p* < 0.05) of AD classified in the stages on the relative abundance of the selected VOCs. Tukey’s test: * *p* < 0.05 and ** *p* < 0.01.

**Figure 6 ijms-24-00707-f006:**
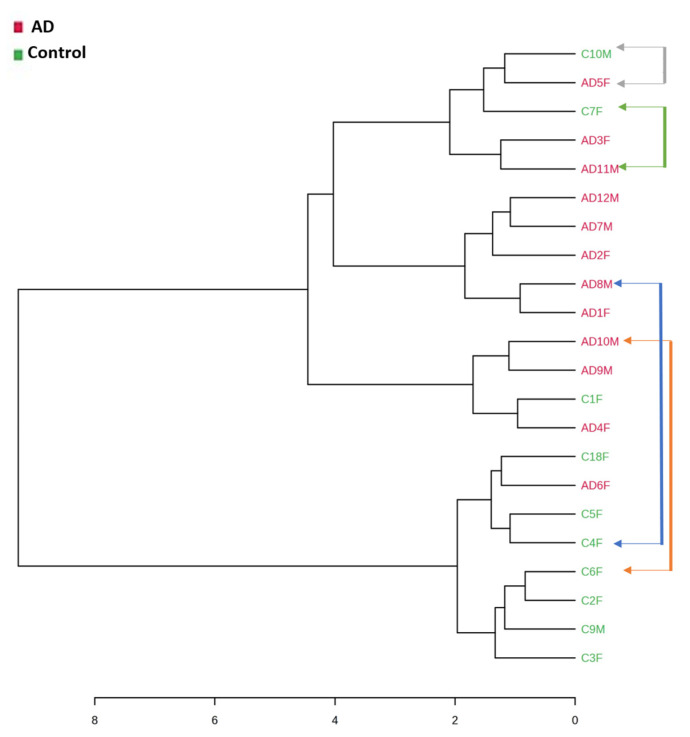
Dendrogram with a Spearman distance measure and a Ward clustering algorithm obtained from the total of VOCs identified in the feces from the control subjects and the Alzheimer’s disease (AD) patients. The pairs of the samples highlighted with an arrow correspond to the married couples; (n = 10, control subjects; n = 12, AD patients).

**Figure 7 ijms-24-00707-f007:**
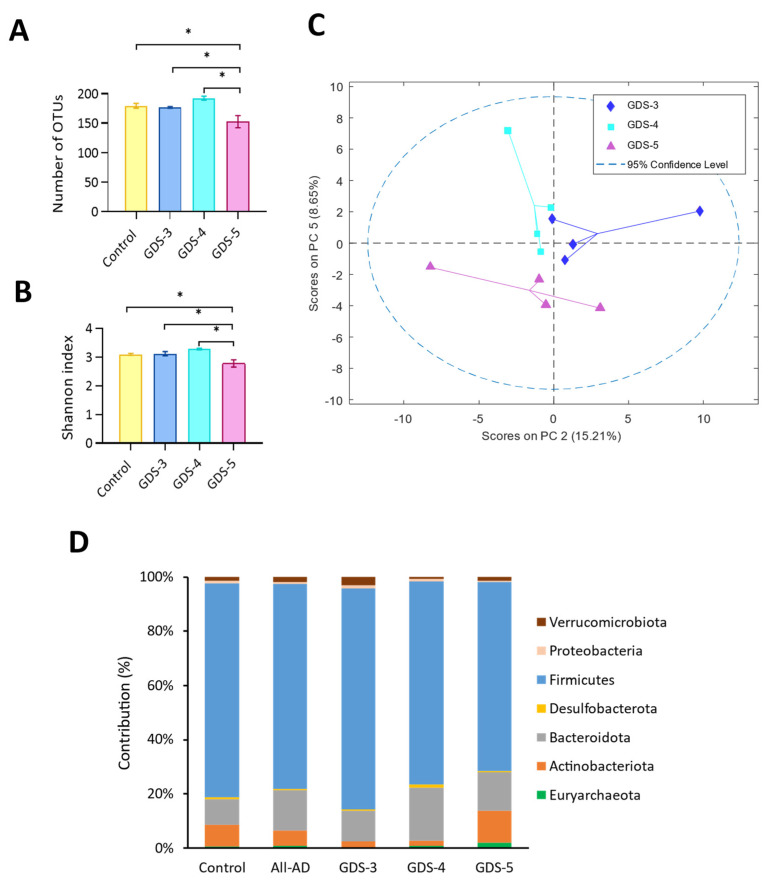
Gut microbiota composition in the feces from the control subjects and the Alzheimer’s disease (AD) patients classified in the stages GDS-3, GDS-4 and GDS-5. (**A**) The richness calculated as the number of the unique operational taxonomic units (OTUs) present in a subject. (**B**) The alpha diversity calculated using the Shannon index at the genus level. The ANOVA shows the significant differences in AD at GDS-5 (*p* < 0.05). The data are the means ± the SEM; Tukey’s test: * *p* < 0.05. (**C**) The principal component analysis (PCA) of the gut microbiota. The scores of the PCA model were carried out with the area of the relative abundance at the genus level from the AD patients at GDS-3, GDS-4 and GDS-5. (**D**) The relative contribution of the bacterial taxa at the phylum level in the control subjects and the AD patients at GDS-3, GDS-4 and GDS-5; (n = 10, control subjects; n = 12, AD patients; 4 per stage).

**Figure 8 ijms-24-00707-f008:**
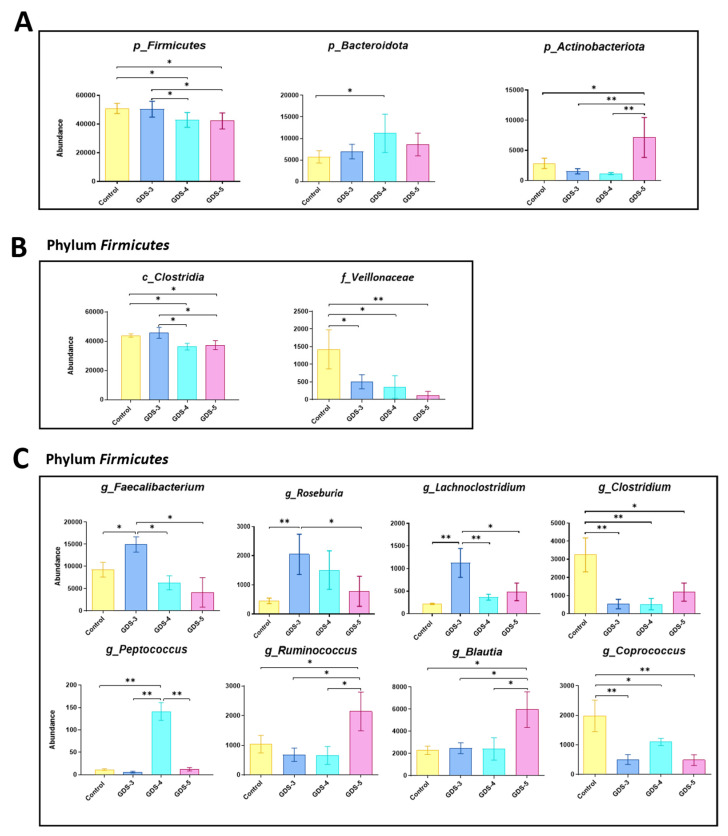
Gut microbiota in the Alzheimer’s disease (AD) patients and the control subjects. The relative abundance of the bacterial taxa in the feces from the control subjects and the AD patients classified in the stages GDS-3, GDS-4 and GDS-5. The data are the means ± the SEM; (n = 10, control subjects; n = 12, AD patients; 4 per stage). (**A**) The bacterial phylum. (**B**) The bacterial class and the family belonging to the *Firmicutes* phylum. (**C**) The genera belonging to the *Firmicutes* phylum. The ANOVA shows an effect (*p* < 0.05) between AD classified in the stages and the bacterial taxa. Tukey’s test: * *p* < 0.05 and ** *p* < 0.01. p, phylum; c, class; f, family; g, genus.

**Figure 9 ijms-24-00707-f009:**
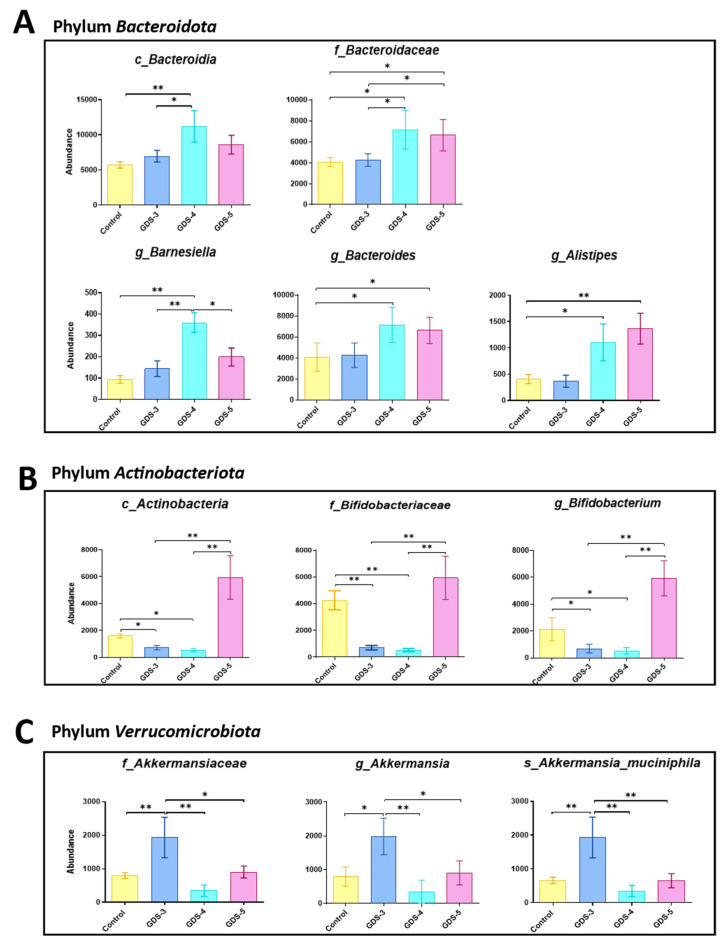
Gut microbiota in the Alzheimer’s disease (AD) patients and the control subjects. The relative abundance of the bacterial taxa in the feces from the control subjects and the AD patients classified in the stages GDS-3, GDS-4 and GDS-5. The data are the means ± the SEM; (n = 10, control subjects; n = 12, AD patients; 4 per stage). (**A**) The bacterial class, family and genera belonging to the phylum *Bacteroidota*. (**B**) The bacterial class, family and genera belonging to the phylum *Actinobacteria*. (**C**) The family *Akkermansiaceae*, genus *Akkermansia* and species *Akkermansia muciniphila*. The ANOVA shows an effect (*p* < 0.05) between the bacterial taxa and the AD patients classified in the stages. Tukey’s test: * *p* < 0.05and ** *p* < 0.01. c, class; f, family; g, genus, s, species.

**Figure 10 ijms-24-00707-f010:**
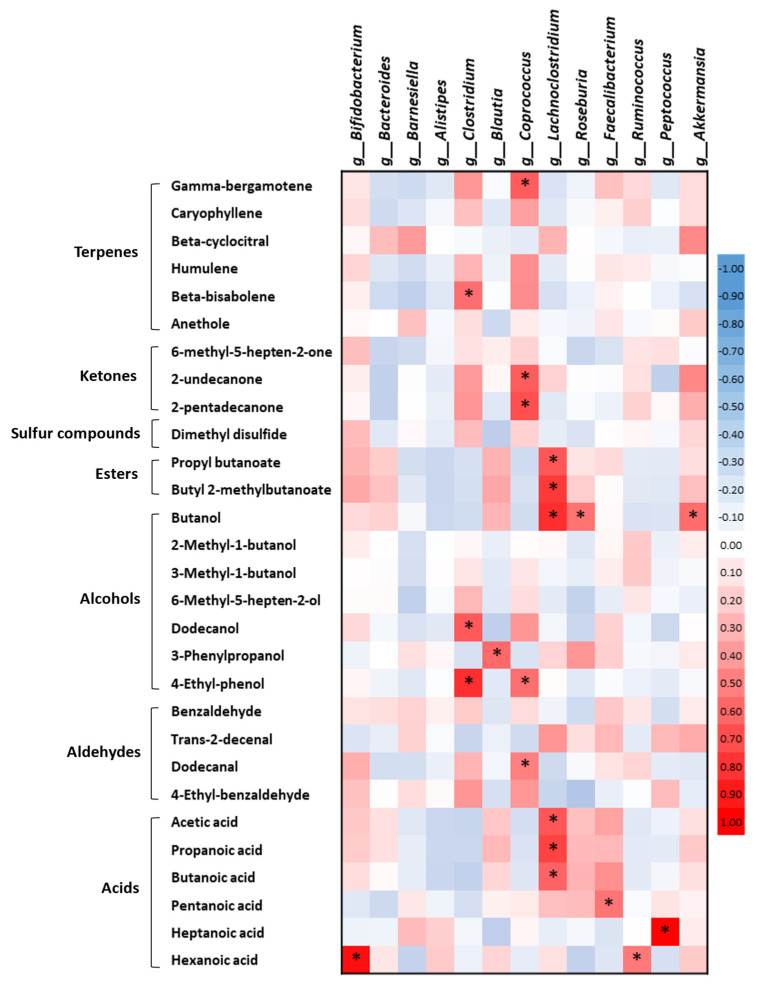
Associations of the fecal volatile organic compounds (VOCs) with the gut microbiota in Alzheimer’s disease (AD). A hierarchical heat map with the correlations between the fecal VOCs selected by the VIP score and the bacterial genera that were statistically significant in the ANOVA analysis. The red squares indicate a positive correlation while the blue squares indicate a negative correlation. The statistical analysis of the correlation coefficient was performed with a t-test. An asterisk indicates the statistical significance; (n = 10, control subjects; n = 12, AD patients).

**Figure 11 ijms-24-00707-f011:**
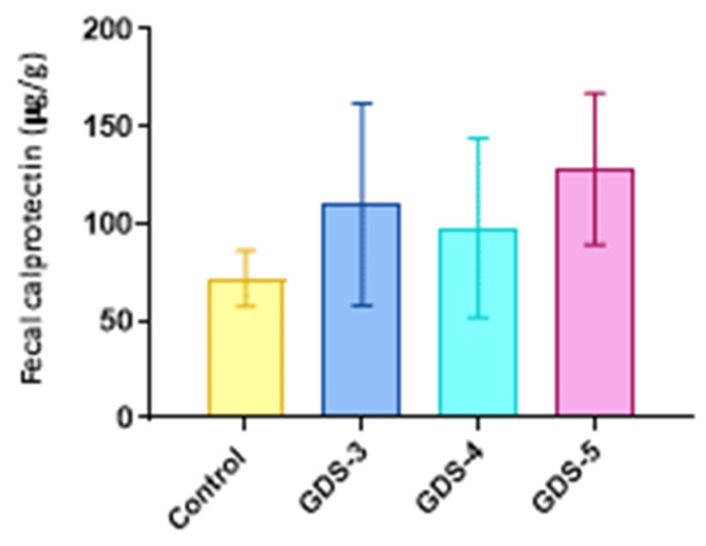
Calprotectin in the Alzheimer’s disease (AD) patients and the control subjects. The fecal samples from the control subjects and the AD patients classified in the stages GDS-3, GDS-4 and GDS-5. The fecal calprotectin levels were expressed as μg calprotectin/g of the feces. The data are the means ± the SEM; (n = 10, control subjects; n = 12, AD patients; 4 per stage).

## Data Availability

The dataset used and/or analyzed during the current study is available from the corresponding author on reasonable request.

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
