# Peer review of "Fecal Volatile Organic Compounds and Microbiota Associated with the Progression of Cognitive Impairment in Alzheimer’s Disease"

_ijms, 2022, doi:10.3390/ijms24010707_

Round 1

Reviewer 1 Report

Cristina Ubeda and colleagues present an interesting manuscript that reports on changes in fecal volatile organic compounds in combination with the changes obtained for the gut microbiome in differently affected Alzheimer's disease patients in comparison to healthy controls. In general, this manuscript sheds new light on the question of contribution of gut microbiota to onset or progression of the disease and thus, will be of high interest to the reader. Besides some minor points, the soundness of the provided work suffers from two facts: the limited number of patients and the unbalanced male/female ratio in the control group. I guess, the data analysis has to somehow take into account that while the Alzheimer population is well-balanced, the control group has a high (80%) female number. Can the authors control for this? At least, this and the small n number have to explicitely be adressed in the discussion but erasing the bias mathematically wold be favorable.

Other points:

1) There are some small language issues that need to be adressed such as (line 60): several reports instead of several reporters. Some commas are missing (e.g., in enumerations). Some wording needs to be corrected (e.g., feces of AD, line 446).

2) Why do the authors connect Blautia and inflammation? There are some reports that even discuss Blautia as a new probiotic (for example, Liu et al., Gut Micerobes 2021).

3) Were the amount of volatile compounds somehow normalized? I guess one normalization was done to the added control, which indicates efffectiveness of the sample preparation. Was there additionally another normalization procedure used to account for e.g. water contetn or any other parameter of the individual samples?

4) To me, the VIP score calculation is not fully clear. Were all compounds with a score higher than 1 considered? If so, why does Figure 4A show 25 and Figure 10 show 27 such compounds? 

5) To my understanding, the wording "the healthy subjects (represented in blue)..." (line 164) is somehow wrong as the volcano plot (Figure 4 B) presents elevated and decreased compounds. In this regard: it is unusual to use a threshhold of 0.1 -  why not 0.05?. Additionally, why has a t-test been used here. There needs to be an adjustment for mutliple testing. Additionally, the auhtors should comment on the fact that the two statistical approaches used here, come to different conclusions. This should also guide them for the subsequent analyses. Moreover, beta-cyclocitral is colered but seems to ly beneath the threshhold.

6) How did the authors select the compounds shown in Figure 5? In general, all can be shown (e.g., also in a Suppl. table) or they may chose from the ones "surviving" both statistical approaches. However, there needs to be a rationale. I wonder why the data for the compound with the best p-value (dodecanol) or the one with the extremest fold change (beta-bisabolene) are both missing.

7) The authors write that mean+/- SEM is shown, but only +SEM is indicated.

8) The authors use a cladogram to demonstrate that eating habits are not the main drivers of the outcome of the analysis (Figure 6). However, as controls and Alzheimer patients are mixed throughout in the cladogram, does this not implicate that also disease is not the main driver?

9) Why are the controls not integrated in Figure 7C?

10) I wonder how the data can get statistically significant at some instances. E.g., Ruminococcus (Figure 8C) or Bifidobacterium (Figure 9B). The SEM are really high and the n numbers very low (n=4 for AD subgroups). Especially for a multiple comparison, p-value adjusted calculation this seems not really plausible.

11) The last sentence of chapter 2.4 (line 287) needs to be rephrased. The authors cannot conclude that the alterations of the gut microbiota are depending on the cognitive benefit. They just see coincidence...

12) How is the heat map in Figure 10 sorted? I also wonder how the compounds and bacterial groups here have been selected. E.g., Bifidobacteria only were significantly different between GDS 5 and 4 (if so).

13) Figure 11 A seems a bit obsolete: 25% out of 4 means 1 (!) patient. Presenting the data like this is somehow misleading. Moreover, I would not interpret the data as an increase if there is no statistical significance. The authors should insert the respective p-value.

14) The discussion seems a bit confusing as preclinical and clinical studies are mixed. Maybe a better separation might help (for example regarding SCFA studies). It would be advantagous to add information about the respective AD models, sex, age of animals and also patient numbers of human studies.

15) The authors might be a bit more careful with usage of references: ref 32. reports on a plant nematode...I would not use it as a reference for the condition withtin gut of animals or human beings. The same holds true for parts of the introduction. some reviews are cited withour giving insight into facts.

16) How was "being cognitively normal" assessed in the controls?

17) Diagnostic tools were quite diverse (biomarkers, ApoE genotype, PET). This might be another limitation.

18) How could "morning fecal sample" be achieved in humans? I guess there is quite a big diversity in timing between individuals...

19) It would be nice to explain the color code within Suppl. Table S1.

Author Response

Cristina Ubeda and colleagues present an interesting manuscript that reports on changes in fecal volatile organic compounds in combination with the changes obtained for the gut microbiome in differently affected Alzheimer's disease patients in comparison to healthy controls. In general, this manuscript sheds new light on the question of contribution of gut microbiota to onset or progression of the disease and thus, will be of high interest to the reader. Besides some minor points, the soundness of the provided work suffers from two facts: the limited number of patients and the unbalanced male/female ratio in the control group. I guess, the data analysis has to somehow take into account that while the Alzheimer population is well-balanced, the control group has a high (80%) female number. Can the authors control for this? At least, this and the small n number have to explicitely be adressed in the discussion but erasing the bias mathematically wold be favorable.

Answer: We are aware of the limited number of AD patients and of the higher number of females in the control group. On the one hand, it is complicated the collection of feces from AD patients that are very special patients and more if we wanted patients at different GDSs. On the other hand, it was a social casuistry that we had more female volunteers within control subjects. We have addressed these limitations of the study in the discussion of the revised version of the manuscript. This study is a first approach, a collaboration between several groups that are working in different fields and because the results were promising and can help elucidate the relationship between the changes in the volatile compounds and those of the gut microbiota that occur in the feces with the AD progression, we wanted to publish them as soon as possible. Further studies in large cohorts are required to obtain more significant associations for each stage of AD.

Other points:

1) There are some small language issues that need to be adressed such as (line 60): several reports instead of several reporters. Some commas are missing (e.g., in enumerations). Some wording needs to be corrected (e.g., feces of AD, line 446).

Answer: Thank you for the observations. The manuscript has been revised and corrected.

2) Why do the authors connect Blautia and inflammation? There are some reports that even discuss Blautia as a new probiotic (for example, Liu et al., Gut Micerobes 2021).

Answer: Thank you for the suggestion. We have mentioned in the discussion a study showing that Blautia was elevated in chronic intestinal inflammation (Forbes et al 2018, PMID: 30545401). There are several associations of Blautia with human diseases including IBD but with discrepancies and most of the properties of this genus are being related with its potential probiotic functions. We now have discussed this part with more cautious in the revised version of the manuscript and included this reference recommended by the reviewer.

3) Were the amount of volatile compounds somehow normalized? I guess one normalization was done to the added control, which indicates efffectiveness of the sample preparation. Was there additionally another normalization procedure used to account for e.g. water contetn or any other parameter of the individual samples?

Answer: The areas were normalized by dividing the peak area of the target ion of each compound by the peak area of the target ion of the internal standard (4-methyl-2-pentanol), expressing them in relative peak area. Moreover, this dataset with the relative areas was normalized by autoscaling prior to any statistical analysis (PLS-DA, PCA, heatmap, etc.). We have indicated this in the Methods section 4.2.

The samples of feces were not pre-treated because the aim of this work was not quantitative, being a first approximation of a rapid qualitative screening of a sample for the onset of Alzheimer's disease. In addition, the samples of feces had similar consistency. 

4) To me, the VIP score calculation is not fully clear. Were all compounds with a score higher than 1 considered? If so, why does Figure 4A show 25 and Figure 10 show 27 such compounds?

Answer: Variable Importance in Projection (VIP) scores estimate the importance of each variable in the projection used in a PLS-DA model and are often used for variable selection. A variable with a VIP Score close to or greater than 1 (one) can be considered important in given model. Variables with VIP scores significantly less than 1 (one) are less important and might be good candidates for exclusion from the model. In the original version of the manuscript, Figure 4A showed a heatmap of the 25 most relevant VOCs for the differentiation according to Pearson correlation coefficient, they were not the VIP variables, due to the statistics for the calculation are different. However, in order to be consistent with the number of volatile compounds selected through the manuscript as relevant for the differentiation, we now have made the Heatmap of Figure 4A with the 29 VIP volatile compounds.

The variables included in Figure 10 are the 29 VOCs selected with VIP value>1 by the PLS-DA, that are also showed in Figure S1c. of the supplementary material, which are marked in purple in the loadings plot of the PLS-DA model. This error has been corrected in figure 10 as two VIP VOCs were missing.

Thank you for the suggestions. All the text related to these figures has been rewritten to be consistent with new figures.

5) To my understanding, the wording "the healthy subjects (represented in blue)..." (line 164) is somehow wrong as the volcano plot (Figure 4 B) presents elevated and decreased compounds. In this regard: it is unusual to use a threshhold of 0.1 -  why not 0.05?. Additionally, why has a t-test been used here. There needs to be an adjustment for mutliple testing. Additionally, the auhtors should comment on the fact that the two statistical approaches used here, come to different conclusions. This should also guide them for the subsequent analyses. Moreover, beta-cyclocitral is colered but seems to ly beneath the threshhold.

Answer: Figure 4B has been corrected in order to use a more common threshold (0.05) according to the reviewer suggestion. Moreover, the text in the manuscript has been rewritten to clarify this statistical analysis and to avoid misleading. The VIP calculation and the volcano plot are mathematically different, and that is the reason why the selected VOCs as important are not exactly the same. Furthermore, the Volcano analysis is univariate, while the selection of VIPs is performed by a multivariate analysis. As VIP variables were more in number, and they contained the variables showed at the volcano plot, the total of them (29) were selected to be studied in the following sections (ANOVA and correlations Figure 10).

6) How did the authors select the compounds shown in Figure 5? In general, all can be shown (e.g., also in a Suppl. table) or they may chose from the ones "surviving" both statistical approaches. However, there needs to be a rationale. I wonder why the data for the compound with the best p-value (dodecanol) or the one with the extremest fold change (beta-bisabolene) are both missing.

Answer: Only those among the 29 VOCs selected as VIPs that showed significant difference in the ANOVA comparing AD patients grouped by GDS were included in Figure 5. On the other hand, we have revised the statistical analysis and found significant changes for dodecanol but not for beta-bisabolene, thus, we have modified the text to clarify all these aspects.

7) The authors write that mean+/- SEM is shown, but only +SEM is indicated.

Answer: Thank you for the suggestion. We now showed both in all the affected figures.

8) The authors use a cladogram to demonstrate that eating habits are not the main drivers of the outcome of the analysis (Figure 6). However, as controls and Alzheimer patients are mixed throughout in the cladogram, does this not implicate that also disease is not the main driver?

Answer: Thank you for the suggestion, we agree with the reviewer’s comment. We have rebuilt the cluster with another algorithm to obtain logical results. In the revised version of the manuscript, we have changed from Euclidean distance measure to Spearman (with Ward clustering algorithm), and now there is a slightly cluster of samples according to the presence or absence of the disease.

9) Why are the controls not integrated in Figure 7C?

Answer: Control subject samples were not included in the PCA showed in Figure 7C to see only the separation between the stages (GDS) of AD, that was the aim to be showed in this figure, it was clearest performing a PCA focusing only on the AD patients, as occurs with the volatile dataset in Figure 3.

10) I wonder how the data can get statistically significant at some instances. E.g., Ruminococcus (Figure 8C) or Bifidobacterium (Figure 9B). The SEM are really high and the n numbers very low (n=4 for AD subgroups). Especially for a multiple comparison, p-value adjusted calculation this seems not really plausible.

Answer: We have revised ANOVA analysis and some SEMs were wrong, sorry about that, we have included in the revised version of the manuscript the corrected figures.

11) The last sentence of chapter 2.4 (line 287) needs to be rephrased. The authors cannot conclude that the alterations of the gut microbiota are depending on the cognitive benefit. They just see coincidence...

Answer: Thank you for the suggestion, this line 287 has been changed being more cautious in the conclusions in the revised version of the manuscript.

12) How is the heat map in Figure 10 sorted? I also wonder how the compounds and bacterial groups here have been selected. E.g., Bifidobacteria only were significantly different between GDS 5 and 4 (if so).

Answer: Volatile compounds were sorted according to the chemical families of the 29 selected VOCs according to the VIP values, and now we indicated the families in the new figure 10. The bacteria were the 13 bacterial genera that were significant in the ANOVA analysis and included the comparisons between AD patients grouped in GDSs or between controls and the AD patients grouped in GDSs. Now, it has been indicated in the revised version of the manuscript.

13) Figure 11 A seems a bit obsolete: 25% out of 4 means 1 (!) patient. Presenting the data like this is somehow misleading. Moreover, I would not interpret the data as an increase if there is no statistical significance. The authors should insert the respective p-value.

Answer: Thank you for the suggestion.  Figure 11A has been remove showing only the figure 11B and revised the results and discussion in the revised version of the manuscript.

14) The discussion seems a bit confusing as preclinical and clinical studies are mixed. Maybe a better separation might help (for example regarding SCFA studies). It would be advantagous to add information about the respective AD models, sex, age of animals and also patient numbers of human studies.

Answer: Thank you for the suggestion. We have included a table with this information in the revised version of the manuscript.

15) The authors might be a bit more careful with usage of references: ref 32. reports on a plant nematode...I would not use it as a reference for the condition withtin gut of animals or human beings. The same holds true for parts of the introduction. some reviews are cited withour giving insight into facts.

Answer: Thank you for the suggestion. We have clarified this information related to reference number 32 in the revised version of the manuscript.

16) How was "being cognitively normal" assessed in the controls?

Answer: control subjects were cognitively normal as indicated by the evaluation carried out in parallel with the AD patients. To clarify this, it has been included in the Methods section 4.1.

17) Diagnostic tools were quite diverse (biomarkers, ApoE genotype, PET). This might be another limitation.

Answer: We are aware of this. For our study, AD patients were classified by cognitive deficit with the evaluation of GDS scale, thus, the groups of AD patients with same GDS were quite homogeneous.

18) How could "morning fecal sample" be achieved in humans? I guess there is quite a big diversity in timing between individuals...

Answer: Since colon motility follows a circadian rhythmic with reduced activity at night and increased in the day and frequently following awaking. For most subjects, defecation reflex is produced in the morning, then the feces can be collected round this time, and it is a good point to get homogeneous samples. We requested it and most of the volunteers of our study let us know that feces were collected first in the morning.

19) It would be nice to explain the color code within Suppl. Table S1.

Answer: Each color distinguished a chemical family. To clarify better we now have indicated with another way.

Reviewer 2 Report

The findings of your research are very intersting by establishing a correlation between non volatile acids and intestinal microbiota and their influence on the clinical stage of alzheimer disease. the metodology of the article is extensivly discussed with use of modern technologies such as phase microextraction/gas chromatography/mass spectrometry (HS-SPME/GC/MS) or fecal microbiom anaylsis using rRNA. The results of the study are illustrated in  numerous graphics, i appreciate the corelations that you established that are worth to be explored by the scientific community. The discussion chapter can be improved please analyze the role of oral health on the influence of intestinal flora in elderly. You can consult: Janto M, Access of Elderly Patients to Oral Health Services and Methods to Improve Oral Health: A Narrative Review. J Pers Med. 2022 Feb 28;12(3):372. .

Please discuss the potential role of this volatile acids to produce intestinal inflammation, permeability and passage of alimentary toxic substances such as heavy metals in general circulation. Please check: Highlighting the Relevance of Gut Microbiota Manipulation in Inflammatory Bowel Disease. Diagnostics (Basel). 2021 Jun 15;11(6):1090. and Microbiome in cancer: Role in carcinogenesis and impact in therapeutic strategies. Biomed Pharmacother. 2022 May;149:112898. doi: 10.1016/j.biopha.2022.112898.

Please discuss the potential use of GCMS technology to determine the blood values of this compound this would be realy helpful as the plasma values are important in reaching general circulation and passing the blood brain barrier. Please check: Mihaela Simona Popoviciu, Cosmin Vesa, Aurora Jurca, Claudia Jurca, Loredana Popa, Dana Zaha, Gabriela Ceavoi, Alexandru Jurca, Farmacia, 2021, 69(2), pp.332-340

Author Response

The findings of your research are very intersting by establishing a correlation between non volatile acids and intestinal microbiota and their influence on the clinical stage of alzheimer disease. the metodology of the article is extensivly discussed with use of modern technologies such as phase microextraction/gas chromatography/mass spectrometry (HS-SPME/GC/MS) or fecal microbiom anaylsis using rRNA. The results of the study are illustrated in  numerous graphics, i appreciate the corelations that you established that are worth to be explored by the scientific community. The discussion chapter can be improved please analyze the role of oral health on the influence of intestinal flora in elderly. You can consult: Janto M, Access of Elderly Patients to Oral Health Services and Methods to Improve Oral Health: A Narrative Review. J Pers Med. 2022 Feb 28;12(3):372.

Answer: Thank you for the suggestion. We have discussed these points and included the references suggested by the reviewer in the revised version of the manuscript.

Please discuss the potential role of this volatile acids to produce intestinal inflammation, permeability and passage of alimentary toxic substances such as heavy metals in general circulation. Please check: Highlighting the Relevance of Gut Microbiota Manipulation in Inflammatory Bowel Disease. Diagnostics (Basel). 2021 Jun 15;11(6):1090. and Microbiome in cancer: Role in carcinogenesis and impact in therapeutic strategies. Biomed Pharmacother. 2022 May;149:112898. doi: 10.1016/j.biopha.2022.112898.

Answer: We now have discussed the potential role of VOCs related to these topics pointed out by the reviewer and included the references suggested by the reviewer in the revised version of the manuscript.

Please discuss the potential use of GCMS technology to determine the blood values of this compound this would be realy helpful as the plasma values are important in reaching general circulation and passing the blood brain barrier. Please check: Mihaela Simona Popoviciu, Cosmin Vesa, Aurora Jurca, Claudia Jurca, Loredana Popa, Dana Zaha, Gabriela Ceavoi, Alexandru Jurca, Farmacia, 2021, 69(2), pp.332-340

Answer: Thank you for the suggestion. We agree with the reviewer that it is a very good idea if we could measure values in the plasma of volatiles compounds. However, there are very scarce works that show the use of GCMS (gas chromatography mass spectrometry) to monitor VOCs in plasma or blood. Plasma is a sample with many limitations for that, is not a good sample to determine volatile compounds with GCMS (gas chromatography mass spectrometry).

On the other hand, as far as we know, we believe that in the paper recommended by the reviewer did not use GCMS but it another technique different completely, it is CGMS that is Continuous Glucose Monitoring System and does not cover the aspects studied in our work.

Reviewer 3 Report

The article describes the organic compounds and microbiota composittion associated with the different stages of Alzheimer´s Disease. Results are interesting and authors have performed a deep characterization. However I have some concerns:

1.- Statistical analysis in Fig 5, 7a, 7b, 8 an 9 is difficult to read. It would be easier to put * pvalue <0.05, ** pvalue<0.01, *** pavelue<0.001. The combination of symbols and letters is difficult to understand. Please clarify this point. Authors also can use lines to join the conditions that are analyzed (for expample control and GDS 5) and add the number of corresponding asterisks.

2.- Authors have use 8 women and only 2 men in control group. please clarify why they have used different number of women and men in the study. Have they identified any differences in organic compounds or microbiota between women and men?

3.- The medication is different between the different GDS groups. Indeed, control group have not received donepezil. Differnet medication intake may produce differneces in microbiota composition. It is a limitation of the study and should be remarked and discussed.

Author Response

The article describes the organic compounds and microbiota composittion associated with the different stages of Alzheimer´s Disease. Results are interesting and authors have performed a deep characterization. However I have some concerns:

1.- Statistical analysis in Fig 5, 7a, 7b, 8 an 9 is difficult to read. It would be easier to put * pvalue <0.05, ** pvalue<0.01, *** pavelue<0.001. The combination of symbols and letters is difficult to understand. Please clarify this point. Authors also can use lines to join the conditions that are analyzed (for expample control and GDS 5) and add the number of corresponding asterisks.

Answer: Thank you for the suggestions, we have revised the statistical analysis and to clarify changed the symbols, also included lines to compare the different groups in the new version of the revised figures.

2.- Authors have use 8 women and only 2 men in control group. please clarify why they have used different number of women and men in the study. Have they identified any differences in organic compounds or microbiota between women and men?

Answer: We did not find any differences in VOCs or microbiota between women and men, but with only two men it cannot be compared. We are aware of the higher number of women in the control group. It was a social casuistry that we had more female volunteers within control subjects. We have addressed this limitation of the study in the discussion of the revised version of the manuscript.

3.- The medication is different between the different GDS groups. Indeed, control group have not received donepezil. Differnet medication intake may produce differneces in microbiota composition. It is a limitation of the study and should be remarked and discussed.

Answer: We are aware of this limitation. Medication was different in AD patient group according to the cognitive deficit, but medication was even different within same AD patient group, thus, for instance half of GDS-3 patients did not receive any medication. Then, at least in the separations it cannot be due to medications, and the dendrogram showed in results did not cluster by stages.  We have addressed this point and included as limitation at the end of the discussion, in the revised version of the manuscript.

It would be very interesting a study controlling all the factors, but we guess it is hard to get all patients without medication, similar diets, life style, etc.

Round 2

Reviewer 1 Report

The authors have appropriately considered my remarks and questions. I would like to congrat them for the highly innovative manuscript.

Reviewer 3 Report

Mnuscript has been improved and it is suitable for publication in this Journal